# Postnatal Outcomes of Fetuses with Prenatal Diagnosis of 6–9.9 mm Pyelectasis

**DOI:** 10.3390/children10020407

**Published:** 2023-02-19

**Authors:** Sivan Farladansky-Gershnabel, Hadar Gluska, Segev Meyer, Maya Sharon-Weiner, Hanoch Schreiber, Shmuel Arnon, Ofer Markovitch

**Affiliations:** 1Department of Obstetrics and Gynecology, Meir Medical Center, Kfar Saba 4428164, Israel; 2Sackler School of Medicine, Tel Aviv University, Tel Aviv 6997801, Israel; 3Department of Neonatology, Meir Medical Center, Kfar Saba 4428164, Israel

**Keywords:** pyelectasis, fetal hydronephrosis, neonatal outcome, long term outcome

## Abstract

Pyelectasis, also known as renal pelvic dilatation or hydronephrosis, is frequently found on fetal ultrasound. This study correlated prenatally-detected, moderate pyelectasis with postnatal outcomes. This retrospective, observational study was conducted at a tertiary medical center in Israel. The study group consisted of 54 fetuses with prenatal diagnosis of pyelectasis on ultrasound scan during the second trimester, defined as anteroposterior renal pelvic diameter (APRPD) 6–9.9 mm. Long-term postnatal outcomes and renal-related sequelae were obtained using medical records and telephone-based questionnaires. The control group included 98 cases with APRPD < 6 mm. Results indicate that fetal pyelectasis 6–9.9 mm was more frequent among males (68.5%) than females (51%, *p* = 0.034). We did not find significant correlations between 6–9.9 mm pyelectasis and other anomalies or chromosomal/genetic disorders. Pyelectasis resolved during the pregnancy in 15/54 (27.8%) cases. There was no change in 17/54 (31.5%) and 22/54 (40.7%) progressed to hydronephrosis Among the study group, 25/54 (46.3%) were diagnosed with neonatal hydronephrosis. There were more cases of renal reflux or renal obstruction in the study group compared to the control group 8/54 (14.8%) vs. 1/98 (1.0%), respectively; *p* = 0.002. In conclusion, most cases of 6–9.9 mm pyelectasis remained stable or resolved spontaneously during pregnancy. There was a higher rate of postnatal renal reflux and renal obstruction in this group; however, most did not require surgical intervention.

## 1. Introduction

Fetal hydronephrosis is a common finding on antenatal ultrasound. It is found in approximately 1–5% of pregnancies [1,2]. This relatively high incidence reflects advances in ultrasound technology, as well as in clinical experience.

The natural history of fetal hydronephrosis has engendered medical interest and has been investigated by maternal-fetal experts, as well as by pediatric nephrologists and urologists. However, correlations between prenatal and post-natal ultrasound findings, as well as the final renal diagnosis, remain inconclusive. Lack of uniformity in measurement techniques and variations in assessing and categorizing urinary tract dilation are the main reasons for the differences between pre-and postnatal diagnoses.

The work-up needed for fetal hydronephrosis has been a controversial subject for decades. The goal of prenatal follow-up in this field is to detect the cases of fetal hydronephrosis that will require evaluation or intervention to mitigate adverse outcomes, while limiting testing in cases that are due to a benign condition.

When hydronephrosis is identified, the common practice is to thoroughly evaluate the fetus by performing a meticulous anatomy scan ultrasound and a renal-directed sonogram to rule out other malformations, as well as to further investigate potential genetic causes. In addition, prenatal counseling with a pediatric nephrologist is performed.

Furthermore, fetal MRI is usually performed when sonographic findings are confusing, when a more thorough assessment is needed because of multiple anomalies, or when anhydramnios or oligohydramnios is detected [3]. Pyelectasis is included as one of the soft ultrasonographic markers of aneuploidy, in addition to echogenic bowel, echogenic intracardiac focus, choroid plexus cyst, and thickening of the nuchal fold.

There is a lack of consensus regarding the threshold anteroposterior renal pelvic diameter (APRPD) that defines clinically significant fetal hydronephrosis requiring postnatal follow-up and is likely to indicate renal pathology. Previous studies have shown increased risk of significant congenital kidney and urinary tract anomalies in cases of severe fetal hydronephrosis [4,5], as well as greater need for postnatal antibiotic treatment and surgical intervention [6,7]. Nevertheless, data regarding the natural course of mild fetal hydronephrosis are scarce. The condition is also referred to as pyelectasia, and is defined as pelvic measurement of 6–9.9 mm in the third trimester.

The objectives of this study were to evaluate the sonographic, obstetrical, neonatal and pediatric characteristics of fetuses diagnosed with APRPD measuring 6–9.9 mm in the second trimester of pregnancy, and to define the natural course and postnatal outcomes of these cases.

## 2. Materials and Methods

This retrospective, observational study was conducted at a tertiary medical center in Israel. The study group consisted of fetuses with a prenatal diagnosis of mild hydronephrosis on ultrasound scan during the second trimester, defined as APRPD of 6–9.9 mm. The ultrasound scans were conducted by a perinatologist in the Department of Obstetrics and Gynecology at Meir Medical Center, Kfar Saba, Israel, from January 2009 through December 2019.

The control group included fetuses with normal APRPD measured during routine second trimester anatomy scans in the same medical center and time frame.

### 2.1. Study Population

Cases with APRPD 6–9.9 mm (study group) and cases with <6 mm (control group) were detected by searching institutional electronic diagnostic codes. The study group included women of ages 18 to 45 years, whose fetuses presented with pyelectasia identified in the second trimester of gestation and who had at least two ultrasound examinations at our hospital. Cases with multiple, extra-renal malformations, chromosomal or genetic abnormalities, intrauterine fetal death or perinatal death, cases that underwent termination of pregnancy and cases lost to follow-up were excluded.

### 2.2. Sonographic Examination

The anteroposterior (AP) diameter of the renal pelvis, visualized in the transverse plane at the level of the renal hilum was measured according to the standard method used to evaluate the grade of antenatal fetal pyelectasis [8].

The transvaginal probes used were 5–9 MHz Voluson E10 or E8 (GE Healthcare, Chicago, IL, USA). The abdominal probes used were 2–5 MHz Voluson E10 or E8 (GE Healthcare). All exams were performed by the same experienced obstetrical-gynecological sonographers.

### 2.3. Urinary Tract Dilation Classifcation

The renal findings were evaluated by the 2021 update on the urinary tract dilation (UTD) classification system published by Nguyen et al. as presented in Figure 1 [9].

### 2.4. Outcome Data

For all cases identified, clinical obstetric characteristics were abstracted from the electronic medical records, including maternal age at diagnosis, gravity, parity, singleton vs. twin pregnancy, fetal sex, obstetrical history, previous diagnosis of hydronephrosis or pyelectasia, and gestational age at diagnosis. Sonographic findings, including bilateral vs. unilateral pyelectasis, accurate measurements of APRPD and other fetal structural abnormalities (musculoskeletal, cardiac, and central nervous, genitourinary and gastrointestinal systems) were also documented.

Prenatal work-up during pregnancy included fetal karyotype and chromosomal microarray analysis (CMA, chorionic villous sampling and or amniocentesis), MRI, echocardiogram, targeted anomaly scans to the fetal urinary system and amniotic fluid volume assessment. Not all listed tests and studies were performed in all patients. The urinary tract targeted scans included APRPD dynamics and evaluation of the ureters, presence of caliectasis, renal cortical thickness, corticomedullary differentiation and evaluation of the urinary bladder. All ultrasound scans were performed by obstetrical-gynecological sonographers with more than 10 years of experience in fetal ultrasound. Both fetal kidneys were scanned in the sagittal, axial and coronal planes in each ultrasound exam. Data regarding birth weight, incidence of preterm deliveries, mode of delivery and postnatal diagnosis of fetal structural abnormalities were abstracted from the patients’ electronic medical records.

Long-term postnatal outcome data (5 years) and data regarding renal-related sequelae were obtained from the electronic medical records and from telephone-based questionnaires. Information regarding postnatal hydronephrosis, prophylactic antibiotics, urinary tract infection in the first month and after two months of life, reflux or renal obstruction, renal scarring, urine protein, chronic hypertension, and any renal-related surgical interventions, were collected. Cases where medical records or telephone questionnaires were not available were excluded.

### 2.5. Statistical Analysis

Data were analyzed using SPSS-25 software (IBM Corp., Armonk, NY, USA). Statistical significance between two groups was calculated using Chi-square or Fisher’s exact test for differences in quantitative variables and *t*-test or Mann-Whitney for continuous variables, each when appropriate. *p* < 0.05 was set as the level of statistical difference because multiple comparisons were performed.

## 3. Results

During the study period, 79 fetuses met the inclusion criteria and had at least two ultrasound examinations in the Ultrasound Unit at the Obstetrics and Gynecology Department. Among them, 25 cases were lost to follow-up. The study group consisted of 54 fetuses with APRPD f 6–9.9 mm. The control group included 98 cases with APRPD < 6 mm.

Demographic and obstetric characteristics of the study group are summarized in Table 1. Mean maternal age at birth was 30.6 ± 5.3 years and mean gestational age at delivery was 38 ± 2.0 weeks. Most fetuses (35/54, 64.8%) were born vaginally.

Pyelectasis measuring 6–9.9 mm was found more frequently in male than female fetuses (68.5% vs. 51%, respectively; *p* = 0.034).

Sonographic fetal renal parameters and prenatal work-up of the study group are presented in Table 2. Unilateral pyelectasis was diagnosed in 32/54 (59.3%) cases. Pyelectasis resolved during the pregnancy in 15/54 (27.8% cases). There was no change in 17/54 (31.5%) cases, and 22/54 (40.7%) progressed to hydronephrosis. Cortico-medullary differentiation was observed in all study group cases, while megaureter was found in 5/54 (9.3%) and enlarged calyces in 19/54 (35.2%) cases.

Additional anomalies found during prenatal ultrasound included cardiac defects (5/54 (9.3%)) and central nervous system anomalies (2/54 (3.7%)). Fetal echocardiography was performed in 20/54 (37%) fetuses diagnosed during pregnancy.

Genetic counseling was performed in 44/54 (81.5%) cases. Amniocentesis or chorionic villous sampling was performed in 15/54 (27.8%) cases. All of these underwent CMA. All genetic samples yielded normal results, except for one case.

The study group had significantly more male fetuses than the control group (37/54 (68%) vs. 50/98 (51%), respectively; *p* = 0.03) and more cases with a history of pyelectasis or hydronephrosis in previous pregnancies (4/54 (7.4%) vs. 0/98, *p* = 0.03). In addition, pyelectasis was more common in twin pregnancies between groups (9.3% vs. 0, respectively; *p* = 0.005; Table 1). Figure 2 shows an axial plane image of bilateral pyelectasis at 24 weeks of gestation.

Neonatal sequalae of children with pyelectasis is presented in Table 3. Immediate postnatal treatment was multidisciplinary and included pediatrician and pediatric urologist evaluations, further testing, surgery if indicated and long-term follow-up. Among the 54 cases with prenatal diagnosis of 6–9.9 mm pyelectasis, 25 (46.3%) were diagnosed with neonatal hydronephrosis. There were no other significant renal pathologies diagnosed after birth compared to the control group. There were more cases of renal reflux or renal obstruction in the study group compared to the control group 8/54 (14.8%) vs. 1/98 (1.0%), respectively; *p* = 0.002. Renal sonographic scan was performed more often in the study group (15/54 (27.8%) vs. controls (1/98 (1.0%), *p* = 0.001).

There was no significant difference in the rate of surgical procedures between groups during the first 5 years of life (12/54 (22.2%) vs. 17/98 (17.3%), respectively; *p* = 0.464). There was no case of chronic hypertension in either group.

## 4. Discussion

This study evaluated the prenatal and postnatal findings of fetuses with 6–9.9 mm pyelectasis. The prenatal and postnatal data presented in this study are among the largest and most specific series of fetuses diagnosed with 6–9.9 mm pyelectasis on routine sonographic scan in low-risk, pregnant women. Pyelectasis was more frequent in males, unilateral in most cases, and was not involved in increased risk for genetic abnormalities. In most cases, it did not deteriorate into hydronephrosis. Most newborns with prenatal diagnosis of pyelectasis needed prophylactic antibiotic treatment. Reflux or obstructive pathology was diagnosed in 14% of cases; however, most did not require surgical intervention.

Urinary tract anomalies are common findings in prenatal ultrasound due to its complex embryonal development [10,11]. Dilatation of the renal pelvis (pyelectasis) is the most common. The incidence of pyelectasis is one to five cases among every 500 newborns [8,12] and the prevalence is 3% of all pregnancies [10,13]. In many cases, it is diagnosed during routine fetal weight estimation scan performed late in pregnancy. Therefore, it poses a clinical dilemma regarding the follow-up and work-up needed. Moreover, the limited data regarding this finding, as well as the extra exams that are performed during the follow-up, cause an emotional burden on the pregnant woman.

In most studies, renal pelvic dilation is a transient physiologic state that is rarely related to clinically significant pathology However, a small portion of cases of fetal hydronephrosis with pelvic dilatation of 10 mm or more, is caused by congenital nephro-uropathies, such as ureteropelvic junction obstruction, vesicoureteral reflux or others. These malformations require postnatal assessment of the infant urinary system, as well as medical treatment or surgical intervention to preserve renal function. Thus, prenatal ultrasound detection is important to allow prompt postnatal follow-up to identify newborns who need prophylactic antibiotics or invasive intervention [4,14]. It is challenging to differentiate between physiological and pathological dilatation of the renal pelvis in the antenatal period. Unlike hydronephrosis, which is defined as pyelectasis more than 10 mm, the appropriate antenatal follow-up of fetuses with 6–9.9 mm pyelectasis is not well defined.

Various fetal renal grading systems have been proposed in the literature, including descriptive grading (mild, moderate or severe), quantitative (antero-posterior renal pelvis diameter), and semiquantitative, as developed by the Society for Fetal Urology [13]. The urinary tract dilataion (UTD) classification system is a unified method of describing urinary tract dilation in fetuses and infants using common terminology with the purpose of determining which children have obstructive uropathy. It is based on measuring the anterior–posterior renal pelvic diameter (APRPD) which is the maximum intrarenal diameter of the renal pelvis measured in the transverse plane of the kidney, with an exception made for antenatal studies where the intrarenal and extrarenal pelvis are not distinguishable [9].

In a prospective study aimed to investigate the natural course of fetal hydronephrosis detected on routine anomaly scan performed at 18–23 weeks, this finding was diagnosed in 2.3% (268/11,465) of cases. Mild hydronephrosis, defined as APRPD ≥ 4 mm, was found in 80.6% (216/268) of cases, and moderate/severe hydronephrosis, defined as APRPD ≥ 7 was diagnosed in 19.4% (52/268) of cases. The hydronephrosis diagnosed in this study resolved in 88% of fetuses in the antenatal or early neonatal period. None of the fetuses with mild hydronephrosis, and approximately one in three fetuses with moderate-to-severe hydronephrosis required surgery. The authors concluded that while mild fetal hydronephrosis was associated with excellent prognosis, moderate-to-severe fetal hydronephrosis was associated with poorer outcomes and, therefore, should be followed up more carefully during the antenatal and postnatal periods [14].

In contrast to the low rate of postnatal surgical treatment mentioned in this study [14], other studies reported higher rates of surgical intervention, ranging from 25% to 40% in fetuses with hydronephrosis [4].

The underlying etiology of prenatal pyelectasis remains obscure in most cases. Furthermore, the relation between fetal hydronephrosis and postnatal renal pathologies is very challenging for pediatricians due to the lack of substantial data regarding this issue.

To the best of our knowledge, this is one of few studies that provides firm information regarding the natural course of prenatal pyelectasis. We found that most cases with 6–9.9 mm pyelectasis resolve or remain unchanged during pregnancy. However, in 40.7% of cases, the dilatation increased to APRPD > 10 mm during third trimester follow-up ultrasound. Hence, we believe that this finding necessitates prenatal follow-up of the size of the renal pelvis during the third trimester to detect cases that will deteriorate to severe hydronephrosis, which is a completely different clinical condition.

Severe hydronephrosis is caused by congenital anomalies of the kidney and urinary tract. The main pathologies associated with severe hydronephrosis include vesicoureteral reflux, ureteropelvic junction obstruction and ureterovesical junction obstruction. Severe bilateral hydronephrosis may lead to serious complications to the growing fetus, including dysplastic changes in the kidney, oligohydramnios, and pulmonary hypoplasia, as well as renal failure in the neonatal period [15].

Pyelectasis may be the prenatal presentation of several urinary tract abnormalities, ranging from vesico-ureteric reflux to obstructive diseases with different grades of severity. Thus, parental prenatal counseling regarding this finding is challenging for obstetricians.

Recently, Loardi et al. [10] published a study regarding the correlation between prenatal findings and the postnatal urological outcomes of moderate and severe fetal pyelectasis. They retrospectively analyzed 90 cases with renal pelvic diameters 10–25 mm, detected prenatally. The prenatal ultrasound findings were compared to postnatal renal function assessed by plasma creatinine level and/or renal imaging performed before surgery. They concluded that prenatal ultrasound diagnoses of moderate and severe renal pelvic diameter are highly predictive of postnatal renal damage and should be addressed properly in prenatal counseling. These results are consistent with our findings, which focused on APRPD 6–9.9 mm.

Although the prognosis of 6–9.9 mm pyelectasis is generally favorable as compared to severe hydronephrosis, this finding requires careful, specialized follow-up and work-up. In cases where pyelectasis does not change in size or resolves, we suggest routine pregnancy follow-up without further evaluation. After birth, these neonates should be followed-up according to their clinical course. In cases where there the size of the renal pelvic diameter increases to 10 mm or more, a multidisciplinary approach should be undertaken and additional work-up, including genetic counseling and testing, fetal echocardiogram, detailed ultrasound follow-up of the fetal urinary system and amniotic fluid index, and pediatric nephrologist and/or urologist consultation. Postnatal treatment in the latter group should be based on the clinical work up performed after birth.

Corteville et al. [16] indicated that the potential to miss significant neonatal disease was greater when a threshold value of APRPD greater than 10 mm was used, as recommended by Grignon et al. [17]. In contrast to these reports, we used a lower threshold of APRD 6–9.9 mm to diagnose fetal renal pyelectasis and found no significant renal disease in these cases. We found that 27.8% of cases resolved during pregnancy and 53.7% resolved postnatally.

Previous studies aimed to examine the risk for renal abnormalities in fetuses with pyelectasis. Jawson et al. [18] evaluated the postnatal outcomes of fetuses with renal pelvic diameter > 5 mm detected at the 20-week anomaly scan. They reported that vesico-ureteric reflux (VUR) was diagnosed in 23/104 (22%) cases. Interesting, among 14 infants with VUR, the postnatal ultrasound scan was normal, indicating that ultrasound is not the best diagnostic modality for VUR, as 55% of cases did not have an enlarged APRPD. In contrast to our study, this report included cases with APRPD > 10 mm. Ahamed et al. [19] reported that 40% of cases of fetal renal pyelectasis ≥5 mm, resolved postnatally. Compared to these studies, the resolution rate of prenatal pyelectasis was higher in our study. We believe that this difference is because we studied the specific group of fetuses with APRPD 6–9.99 mm RPD, which is a lower risk group.

In accordance with other published series [15] our study also showed a male predominance of 6–9.9 mm fetal pyelectasis of 68.5% compared to 51% among females (*p* = 0.034). Surgery was also more prevalent in male neonates. The male predominance was similar in mild, moderate and severe renal pelvic dilatation.

Early second trimester pyelectasis is considered a soft marker for Down syndrome. Isolated fetal pyelectasis detected on mid-trimester ultrasound is associated with an increased likelihood ratio of 2.78 for trisomy-21, based on maternal age and serum screening tests [20]. This elevated risk should be taken into account, and genetic counseling and testing should be performed to rule out Down syndrome. Without disregarding this important information, the current study focused on the renal and urological outcomes of these fetuses. We did not find significant correlations between 6–9.9 mm pyelectasis and other anomalies or chromosomal/genetic disorders. It is important to inform women of the available screening and diagnostic genetic studies and their detection rates [21]. Previous studies correlated between pyelectasis and chromosomal abnormalities; thus, defining it as a “soft marker” [22]. However, a more recent study questioned this axiom and found that this correlation was true only when pyelectasis is accompanied by other structural or hormonal anomalies [23]. Su et al. [24] found that fetal pyelectasis was the most common renal ultrasound finding, but only 2.37% harbored pathogenic or likely pathogenic copy number variants in isolated cases. Our results support the latter study.

An older study considered unilateral pyelectasis to be associated with increased risk for postnatal renal surgery compared with bilateral pyelectasis [25]. Persistency was associated with unilateral compared with bilateral renal lesions. However, we did not find an association between laterality and the risk for postnatal pathology. This discrepancy may be because our study did not include RPD > 9.9 mm.

### Strengths and Limitations

The strengths of this study are that it included a large, well-defined cohort. All cases were diagnosed by the same obstetrical-gynecological sonographers with uniform examination protocols and followed by the same pediatric urologist in a university-affiliated, tertiary medical center. In addition, this is among the largest reported series of mild-to-moderate cases of fetal pyelectasis with postnatal follow-up and prenatal analysis. Data retrieval from patient files was meticulous, and we were able to obtain information regarding the prenatal work-up and postnatal outcomes of the cases.

The study is not without limitations. Some of which were derived from the retrospective nature of the study. A recall bias is possible, because the study follow-up was retrospective and based on questionnaire data.

## 5. Conclusions

Most cases of fetal renal pyelectasis 6–9.9 mm remained stable or resolved spontaneously in-utero or ex-utero, or improved after birth. However, there was a higher incidence of reflux and/or renal obstruction in this group, although most did not require surgery.

Although we found mild-to-moderate pyelectasis to be associated with good postnatal outcomes, we believe that all cases of fetal renal pyelectasis 6–9.9 mm detected during the mid-second trimester ultrasound should be followed antenatally by ultrasound. In isolated cases, there is no indication for fetal echo, amniocentesis or intrusive neonatal renal testing such as voiding cystourethrogram, as opposed to severe pyelectasis of 10 mm or greater. The findings of this study should be helpful when consulting with parents regarding the follow-up and prognosis of prenatally-diagnosed pyelectasis. Fetuses with persistent pyelectasis should be evaluated after birth and followed closely until resolution to rule out reflux, obstruction or other concerns.

## Figures and Tables

**Figure 1 children-10-00407-f001:**
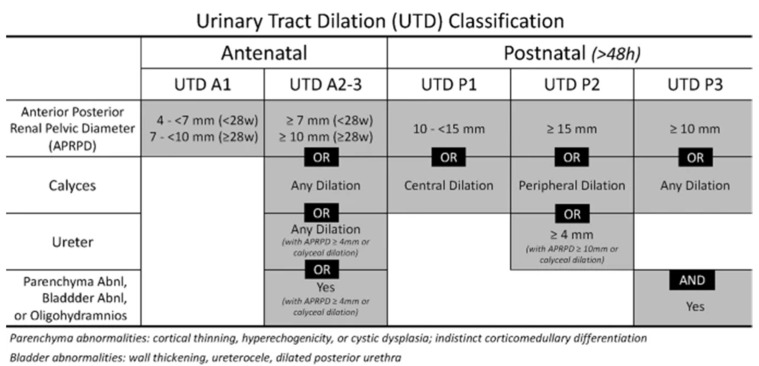
Table adapted from Nguyen, 2022, pp. 740−751 [9].

**Figure 2 children-10-00407-f002:**
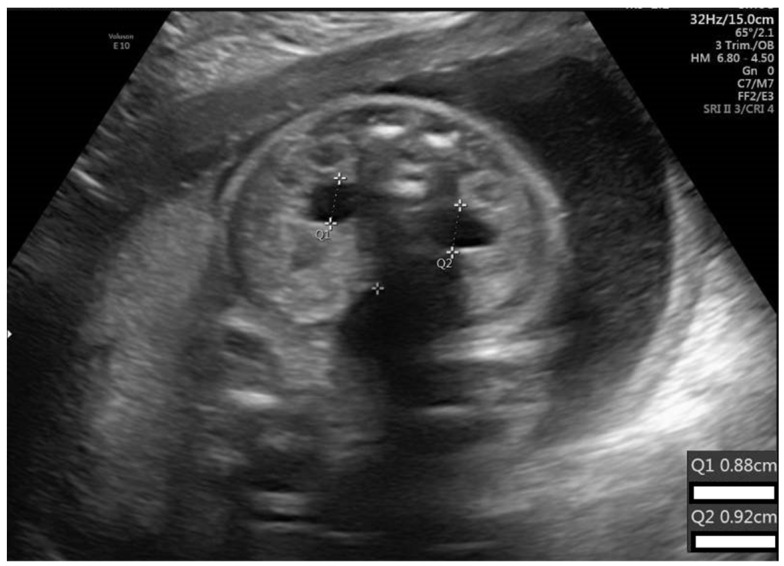
Axial plane image of bilateral pyelectasis at 24 weeks of gestation (8.8 mm, 9.2 mm).

**Table 1 children-10-00407-t001:** Demographic, obstetric and pre-natal work-up of fetuses with pyelectasis vs. control group.

Demographic and Obstetric Characteristics	Pyelectasis (N = 54)	Control Group (N = 98)	*p*-Value
Maternal age at diagnosis, years (mean ± SD)	30.6 ± 5.3	29.9 ± 4.4	0.384
Gravity	2.3 ± 1.2	2.3 ± 1.5	0.628
Parity	1.8 ± 0.8	1.9 ± 1.1	0.869
Male fetus	37/54 (68.5%)	50/98 (51.0%)	0.034
Twin pregnancy	5/54 (9.3%)	0/98 (0%)	0.005
Pyelectasis/hydronephrosis in previous pregnancy	4/54 (7.4%)	0/98 (0%)	0.035
Mode of delivery			
Vaginal	35/54 (64.8%)	71/98 (72.4%)	0.630
Vacuum	4/54 (7.4%)	4/98 (4.1%)	0.642
Cesarean section	15/54 (27.8%)	23/98 (23.5%)	0.527
Delivery at term (>37 weeks)	47/54 (87.0%)	90/98 (91.8%)	0.342
Birth weight	3222.00 ± 623	3134.41 ± 535	0.365
Apgar 1-min	9 ± 0.2	8.9 ± 0.6	0.747
Apgar 5-min	10 ± 0.2	9.9 ± 0.2	0.798
Prenatal workup and diagnosis			
Oligohydramnios	3/54 (5.6%)	3/98 (3.1%)	0.542
Polyhydramnios	2/54 (3.7%)	0/98 (0.0%)	0.471
Other renal findings	1/54 (1.9%)	1/98 (1.0%)	0.693
Genetic counseling	44/54 (81.5%)	76/98 (77.6%)	0.569
Chorionic villus sampling/Amniocentesis	15/54 (27.8%)	33/98 (33.7%)	0.454
Pathologic chromosomal microarray analysis	1/15 (6.6%)	0/33 (0%)	0.610
Echocardiogram	2/54 (3.7%)	3/98 (3.06%)	0.945

Categorical variables results are presented as N (%) and continuous variables as mean ± standard deviation (SD).

**Table 2 children-10-00407-t002:** Fetal ultrasound renal findings and prenatal work-up of the study group.

Characteristics	Value
At first diagnosis	
Gestational age, weeks (mean ± SD)	22.01 ± 8.4
Right pyelectasis size, mm (mean ± SD)	6.71 ± 2.4
Left pyelectasis size, mm (mean ± SD)	7.22 ± 2.0
Right renal cortex size, mm (mean ± SD)	7.01 ± 2.7
Left renal cortex size, mm (mean ± SD)	6.93 ± 2.6
Unilateral Pyelectasis	32/54 (59.3%)
Oligohydramnios	3/54 (5.6%)
Polyhydramnios	2/54 (3.7%)
Megaureter	5/54 (9.3%)
Enlarged calyces	19/54 (35.2%)
Cortico-medullary differentiation	54/54 (100%)
Other renal findings	1/54 (1.9%)
Cardiac anomalies	5/54 (9.3%)
Central nervous system anomalies	2/54 (3.7%)
Genetic counseling	44/54 (81.5%)
Chorionic villus sampling/Amniocentesis	15/54 (27.8%)
Pathologic chromosomal microarray analysis results	1/15 (6.6%)
Further renal examinations	
Progression of hydronephrosis	22/54 (40.7%)
Resolution of pyelectasis	15/54 (27.8%)
No change	17/54 (31.5%)
Cardiac Echo	20/54 (37%)

Categorical variables results are presented as N (%) and continuous variables as mean ± standard deviation (SD).

**Table 3 children-10-00407-t003:** Sequalae of children with pyelectasis vs. control group.

Characteristic	Pyelectasis (N = 54)	Control Group (N = 98)	*p*-Value
Neonatal hydronephrosis	25/54 (46.3%)	0/98 (0%)	0.000
Antibiotics in first year of life	34/54 (63.0%)	0/98 (0%)	0.000
Other pathologies, discovered after birth	5/54 (9.3%)	8/98 (8.2%)	0.817
Urinary tract infection, in the first two months of life	1/54 (1.9%)	0/98 (0%)	0.355
Chronic kidney disease	1/54 (1.9%)	0/98 (0%)	0.355
Reflux/obstruction	8/54 (14.8%)	1/98 (1.0%)	0.002
Renal scanning	15/54 (27.8%)	1/98 (1.0%)	0.001
Renal scaring	1/15 (6.6%)	0/1 (0%)	1.000

Categorical variables results are presented as N (%) and continuous variables as mean ± standard deviation (SD).

## Data Availability

Data are available upon request.

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
