# Peer review of "Postnatal Outcomes of Fetuses with Prenatal Diagnosis of 6–9.9 mm Pyelectasis"

_children, 2023, doi:10.3390/children10020407_

Round 1
Reviewer 1 Report
I think this study warrants publication.
My only suggestion is to carefully check for repetitions and try to remove or at least diminish them. Examples are that the content of Tables are also provided in the accompanion texts. Table 1 is also part of Table 3. Please also check the Discussion for repetitions.
Very minor points: Paragraph 2.3, Prenatal work-up --: ")" is missing in line 92. Line 99: Data. Line 102: What is HTN? Line 120: 3.1? Line 133: What is CVS? Line 139: --- than the "control" group. Finally, reconsider the number of decimals in some of the data, Typicle example: Birth weight 3222.0 ± 623.1 instead of 3222 ± 623.
Author Response
Dear Reviewer, Thank you so much for your important and detailed comments. My only suggestion is to carefully check for repetitions and try to remove or at least diminish them. Examples are that the content of Tables are also provided in the accompanion texts. Table 1 is also part of Table 3. Please also check the Discussion for repetitions. Authors reply: Thank you for your comment. We removed repetitive sentences. We removed the original Table 1. Paragraph 2.3, Prenatal work-up --: ")" is missing in line 92. Line 99: Data. Line 102: What is HTN? Line 120: 3.1? Line 133: What is CVS? Line 139: --- than the "control" group. Reconsider the number of decimals in some of the data Authors reply: All of the above were corrected.Reviewer 2 Report
Dear authors, I've appreciated your research
pyelectasia is a very frequent finding during the anomaly scan,
so the topic is interesting the paper well written the study well conducted
I would like to suggest some minor revisions
1) as you've said there is still no fixed consensus on when to discuss genetic testing, I think that it is of crucial importance to let women become aware of the meaning of each of the potential tests existing for chromosomal and genetic abnormalities detection, even if not always needed in the presence of pylectasia when it becomes significant it is important to mention the available test to rule out chromosomal and genetic abnormalities but more than everything to make sure that the patient has become aware of the goal and the detection rate of each of these test, I would like to suggest you to mention this concept within the discussion because it is important and in addition because the journal is not specifically addressed to a fetal medicine auditorium, I suggest to read and consider to cite PMID: 33111167
2) I would like to suggest to add some images demonstrating fetal pyelectasia
3) I would like to suggest to add a table with a summary of findings
best regards
Author Response
Dear Reviewer, Thank you so much for your important and detailed comments. 1) I think that it is of crucial importance to let women become aware of the meaning of each of the potential tests existing for chromosomal and genetic abnormalities detection, even if not always needed in the presence of pylectasia when it becomes significant it is important to mention the available test to rule out chromosomal and genetic abnormalities but more than everything to make sure that the patient has become aware of the goal and the detection rate of each of these test, I would like to suggest you to mention this concept within the discussion because it is important and in addition because the journal is not specifically addressed to a fetal medicine auditorium, I suggest to read and consider to cite PMID: 33111167 Authors reply: Thank you for your comment. We added the important concept to the discussion and cited the suggested study as reference 22. 2) I would like to suggest to add some images demonstrating fetal pyelectasia. Authors reply: Thank you for you suggestion. We added a representative image.